# Immunological Profile and Markers of Endothelial Dysfunction in Elderly Patients with Cognitive Impairments

**DOI:** 10.3390/ijms25031888

**Published:** 2024-02-04

**Authors:** Nikolay V. Goncharov, Polina I. Popova, Igor V. Kudryavtsev, Alexey S. Golovkin, Irina V. Savitskaya, Piotr P. Avdonin, Ekaterina A. Korf, Natalia G. Voitenko, Daria A. Belinskaia, Maria K. Serebryakova, Natalia V. Matveeva, Natalia O. Gerlakh, Natalia E. Anikievich, Marina A. Gubatenko, Irina A. Dobrylko, Andrey S. Trulioff, Arthur D. Aquino, Richard O. Jenkins, Pavel V. Avdonin

**Affiliations:** 1Research Institute of Hygiene, Occupational Pathology and Human Ecology of the Federal Medical Biological Agency, bld 93 Kuzmolovsky, Leningrad Region 188663, Russia; 2Sechenov Institute of Evolutionary Physiology and Biochemistry of the Russian Academy of Sciences, St. Petersburg 194223, Russia; 3City Polyclinic No.112, St. Petersburg 195427, Russia; 4Institute of Experimental Medicine, St. Petersburg 197022, Russia; 5Almazov National Medical Research Centre, St. Petersburg 197341, Russia; 6Maximilianovskaya City Hospital No.28, St. Petersburg 190000, Russia; 7Koltsov Institute of Developmental Biology of the Russian Academy of Sciences, Moscow 119334, Russia; 8School of Allied Health Sciences, De Montfort University, The Gateway, Leicester LE1 9BH, UK

**Keywords:** aging, immunosenescence, endothelial cells, acute ischemic stroke, chronic cerebral circulation insufficiency, diabetes mellitus, extracellular vesicles

## Abstract

The process of aging is accompanied by a dynamic restructuring of the immune response, a phenomenon known as immunosenescence. Further, damage to the endothelium can be both a cause and a consequence of many diseases, especially in elderly people. The purpose of this study was to carry out immunological and biochemical profiling of elderly people with acute ischemic stroke (AIS), chronic cerebral circulation insufficiency (CCCI), prediabetes or newly diagnosed type II diabetes mellitus (DM), and subcortical ischemic vascular dementia (SIVD). Socio-demographic, lifestyle, and cognitive data were obtained. Biochemical, hematological, and immunological analyses were carried out, and extracellular vesicles (EVs) with endothelial CD markers were assessed. The greatest number of significant deviations from conditionally healthy donors (HDs) of the same age were registered in the SIVD group, a total of 20, of which 12 were specific and six were non-specific but with maximal differences (as compared to the other three groups) from the HDs group. The non-specific deviations were for the MOCA (Montreal Cognitive Impairment Scale), the MMSE (Mini Mental State Examination) and life satisfaction self-assessment scores, a decrease of albumin levels, and ADAMTS13 (a Disintegrin and Metalloproteinase with a Thrombospondin Type 1 motif, member 13) activity, and an increase of the VWF (von Willebrand factor) level. Considering the significant changes in immunological parameters (mostly Th17-like cells) and endothelial CD markers (CD144 and CD34), vascular repair was impaired to the greatest extent in the DM group. The AIS patients showed 12 significant deviations from the HD controls, including three specific to this group. These were high NEFAs (non-esterified fatty acids) and CD31 and CD147 markers of EVs. The lowest number of deviations were registered in the CCCI group, nine in total. There were significant changes from the HD controls with no specifics to this group, and just one non-specific with a maximal difference from the control parameters, which was α1-AGP (alpha 1 acid glycoprotein, orosomucoid). Besides the DM patients, impairments of vascular repair were also registered in the CCCI and AIS patients, with a complete absence of such in patients with dementia (SIVD group). On the other hand, microvascular damage seemed to be maximal in the latter group, considering the biochemical indicators VWF and ADAMTS13. In the DM patients, a maximum immune response was registered, mainly with Th17-like cells. In the CCCI group, the reaction was not as pronounced compared to other groups of patients, which may indicate the initial stages and/or compensatory nature of organic changes (remodeling). At the same time, immunological and biochemical deviations in SIVD patients indicated a persistent remodeling in microvessels, chronic inflammation, and a significant decrease in the anabolic function of the liver and other tissues. The data obtained support two interrelated assumptions. Taking into account the primary biochemical factors that trigger the pathological processes associated with vascular pathology and related diseases, the first assumption is that purine degradation in skeletal muscle may be a major factor in the production of uric acid, followed by its production by non-muscle cells, the main of which are endothelial cells. Another assumption is that therapeutic factors that increase the levels of endothelial progenitor cells may have a therapeutic effect in reducing the risk of cerebrovascular disease and related neurodegenerative diseases.

## 1. Introduction

The process of aging is accompanied by a dynamic restructuring of the immune response, a phenomenon known as immunosenescence, which includes alterations to immune cell profiles and their functional dynamics, changes to T-cell receptor signaling, cytokine network dysregulation, and compromised regulatory T-cell function [1,2]. Further, damage to the endothelium can be both a cause and a consequence of many diseases, especially in elderly people. Currently, the question is not so much about the involvement of endothelial cells (ECs) in the pathogenesis of these diseases, but more about the shared participation of the endothelium in the development of these kinds of diseases, the primary or secondary nature of violations to the morpho-functional status of ECs, and their dependence on genetic and epigenetic factors [3]. The pathogenesis of many diseases of the central and peripheral nervous systems is, in one way or another, associated with disruption of the blood-tissue barrier, including primarily the blood-brain barrier (BBB). At the capillary level, solutes diffuse through the extracellular spaces and enter the blood through the transport systems of ECs and pericytes [4,5]. Small vessels play a key role in the process of brain autoregulation. Among the acute neurological disorders associated with damage to the cerebral vascular endothelium, the most well-known are stroke, epilepsy, and traumatic lesions of the brain and spinal cord. BBB dysfunction is a characteristic feature of both ischemic and hemorrhagic stroke [6,7]. Besides genetically determined factors, in addition to acute environmental factors, all people experience age-related changes in blood vessels over time, which can become an independent cause of neurodegenerative diseases. The main diseases, in terms of scale and worldwide distribution, are senile dementia of the Alzheimer’s type or sporadic Alzheimer’s disease (SAD) which accounts for 85–90% of total patients [8,9,10]. Further, an important emerging avenue in dementia research is understanding the vascular contributors to dementia. Among the potential causes of vascular cognitive impairment are stroke, microinfarction, hypertension, atherosclerosis, blood-brain-barrier dysfunction, and cerebral amyloid angiopathy (CAA) [3]. Sporadic CAA is a highly prevalent small vessel disease in the aging population with potential severe complications including lobar intracerebral hemorrhage (ICH), cognitive impairment, and dementia [11].

Cerebral circulation insufficiency consists of two types: acute, such as acute ischemic stroke (AIS) or transient ischemic attack (TIA), and chronic, such as chronic cerebral circulation insufficiency (CCCI). The former has received great concern, while the latter has not aroused enough attention so far. CCCI may not be an independent disease. It may be a pervasive state of long-term cerebral blood flow insufficiency caused by a variety of etiologies and is considered to be associated with either the occurrence or recurrence of ischemic stroke, vascular cognitive impairment, and the development of vascular dementia, resulting in disability and mortality worldwide [12]. CCCI, which was initially proposed by Japanese scholars in the 1990s, refers to a state of reduction in cerebral blood flow (CBF) below the physiologically required volume, leading to brain dysfunctions, and this state should last for at least 2 months. CCCI can be secondary to a variety of etiologies, with atherosclerosis predominating [13,14]. Clinical studies revealed that the symptoms of CCCI, such as dizziness and headache, were actually reversible once cerebral circulation was improved. On the contrary, persistent CBF reduction, if not corrected, may evoke AIS, TIA, vascular cognitive impairment, or even dementia [12,15]. Therefore, identifying this condition followed by prompt and effective intervention is enormously valuable.

Damaged ECs may cause dysregulated blood flow, and subcortical ischemic vascular dementia (SIVD) can develop as a result [16,17]. The pathogenesis of SIVD is driven by stenosis and the occlusion of small vessels of the brain, with subsequent white matter ischemia and lacunar infarctions in the subcortical structures. Most risk factors for small vessel disease (SVD) can be regarded as risk factors for SIVD and cognitive impairment. Hypertension is one of the best-known risk factors for SVD and SIVD, though hypotension may be an even more dangerous factor [18,19]. Regardless, decreased regional blood flow is a critical factor in the development of SIVD [20], which is an irreversible condition characterized by a progressive decline in cognitive status, an impairment of memory, speech difficulties, and decreased social abilities. Cerebrovascular dysfunction associated with cognitive impairment is an integral attribute of SIVD; beta-amyloid (Aβ) plaques and hyperphosphorylated tau proteins are also found [21]. Early damage to the BBB cannot be visualized in a living person with early signs of SIVD by instrumental methods of analysis, although a certain success has been achieved [17,22]. For example, the differential assessment of subcortical ischemic vascular cognitive impairment, even with no dementia (SIVCIND), was carried out by the combination of radiomics derived from high-resolution T1-weighted imaging of the brain and machine learning [23]. Recent advances in neuroimaging allow the diagnosis of CAA in the absence of a pathological examination. Current imaging-based criteria have a high diagnostic performance in patients presenting with ICH but are more limited in other clinical contexts such as cognitively impaired patients or asymptomatic individuals [11]. Further research is still needed to improve diagnostic accuracy. In most cases of practical medicine, evidence is obtained as a result of genetic and biochemical analysis and, as a rule, in postmortem studies. One of the most well-known biochemical markers of disruption to the integrity of the BBB is the albumin ratio (Qalb), which is the ratio of cerebrospinal fluid albumin levels to serum albumin levels [24,25,26]. Tight junctions between cerebral endothelial cells play an important role in the function of the BBB, and recent studies have demonstrated the essential role of microRNAs in regulating tight junctions [27]. Recently, the brain-derived neurotrophic factor (BDNF) and circulating arginine metabolites were suggested as potential biomarkers of Alzheimer’s disease (AD) [28,29]. Numerous recent studies have indicated that vascular risk factors are associated with various forms of dementia and that most forms of dementia can be considered an extension of vascular disease.

The endothelium can generate a diverse range of vasoactive compounds and signals, most of which act locally to adjust blood flow in a dynamic fashion to match tissue metabolism. Disruption of these vascular signaling processes (e.g., reduced nitric oxide bioavailability) is typically referred to as endothelial dysfunction, which is a recognized risk factor for cardiovascular disease in patients and occurs early in the development and progression of hypertension, atherosclerosis, and tissue ischemia [30,31]. Endothelial dysfunction is also associated with type II diabetes mellitus (DM) and aging, and an increased mechanistic knowledge of the cellular changes contributing to these effects may provide important clues for interventional strategies [32]. The endothelium also serves as the initial site of interaction for immune cells entering tissues in response to damage and facilitates the actions of both the innate and acquired immune systems to interact with the vascular wall. In addition to representing the main cell type responsible for the formation of new blood vessels (i.e., angiogenesis) within the vasculature, the endothelium is also emerging as a source of extracellular vesicles (EVs) or microparticles for the transport of signaling molecules and other cellular materials to nearby, or remote, sites in the body [30]. Advances in liquid-based assays and molecular biology tools are providing novel potential EC-specific biomarkers for the prediction and diagnosis of endothelial dysfunction. Significant associations between clinically useful indices of endothelial dysfunction, mainly brachial artery flow-mediated dilation (FMD) and an increased number of endothelial microparticles (EMPs) and increased levels of endoglin and endocan, as well as reduced levels of irisin, were observed in subjects with one or more traditional risk factors. However, none have yet entered clinical practice [33]. Among the novel circulating biomarkers related to the endothelium are chemokines, and epigenetic and metabolomic biomarkers [34].

Existing achievements do not solve the problem of risk stratification for neurodegenerative diseases and cognitive disorders in the early stages of their development. Even at later stages, the development of cheaper, more accessible, and minimally invasive diagnostic methods is relevant. The purpose of this study was to carry out immunological and biochemical profiling of elderly people (aged 60–78 years according to the WHO classification) with AIS or CCCI, as well as people with prediabetes or untreated DM. The positive control group consisted of patients with severe signs of SIVD. One of the main tasks was to identify markers of endothelial genesis that indicated endothelial dysfunction. To this end, a special feature of our study was the determination of the level of CD markers of extracellular vesicles (microparticles) of endothelial origin, which could enhance the predictive power of the traditional biomarkers used and could serve as a liquid biopsy to understand the vascular pathophysiology [30,33,35,36].

## 2. Results

This study included elderly people (60–78 years old, Table 1) with criteria as follows: (1) healthy donors (HDs, *n* = 23); (2) patients with a history of AIS (*n* = 24); (3) patients with prediabetes or newly diagnosed diabetes mellitus (DM, *n* = 21); and (4) patients with CCCI (*n* = 32); (5) patients with SIVD (*n* = 20). Their socio-demographic and lifestyle scores were calculated and compared, which included age, gender, smoking and alcohol habits, socialization, life satisfaction, features of physical activity, and night sleep. To assess mental status and diagnose cognitive impairment, the MMSE (Mini Mental State Examination) method and the MOCA test (Montreal Cognitive Impairment Scale) were used (Table 1). 

Using the Kruskal–Wallis test and Dunn’s multiple comparisons test, no significant differences between the groups according to the age of the men and women and their body mass index (BMI) were revealed. Significant changes relative to the control HD group in all four groups of patients were detected in the mental status indicators MOCA and MMSE, with the greatest decreases of 56% and 52%, respectively, in the SIVD group. Significant differences between the groups were also revealed for socio-demographic and lifestyle scores, but for one of these indicators—life satisfaction—there were decreasing differences in the two groups, CCCI and SIVD (by 25%). The scores on alcohol consumption, socialization, and physical activity were significantly different from the control patients just in one group, SIVD (decrease by 67%, 50%, and 33%, respectively), but there were no other groups with significant changes for alcohol scores, the additional AIS group for physical activity scores, and all three other groups for socialization scores. Smoking scores were significantly decreased by 75% in the SIVD group compared with the AIS group, and night sleep features demonstrated no significant differences between the groups. 

Socialization turned out to be a unique lifestyle parameter since it was significantly decreased in the SIVD group compared to the other groups, despite the fact that none of the three groups showed differences in relation to the control or among themselves. This was not an accident or an error in our study, confirmed by recently published data from a large-scale study on the risk factors for young-onset dementia (YOD) [37]. Social isolation was among 15 factors that were significantly associated with a higher YOD risk.

If the decrease in the level of socialization, physical activity, degree of life satisfaction, and even the intensity of smoking for patients in the SIVD group did not come as a surprise, then a significant decrease in alcohol consumption was a somewhat surprising phenomenon. It is a well-known fact that alcohol use disorder (AUD) is associated with an increased risk of cognitive impairments, Alzheimer’s disease, and dementia, especially vascular dementia [38]. In addition, AUD interacts with comorbidities and increases the risk of cognitive impairment. On the other hand, smoking and alcohol consumption were not associated with vascular risk factors for idiopathic normal-pressure hydrocephalus [39]. According to the recent Maastricht study, the association between total alcohol consumption and microvascular dysfunction (MVD) is non-linear, i.e., J-shaped. Moderate versus light total alcohol consumption is significantly associated with decreased MVD, after full adjustment [40]. The shape of the curve differed with sex, history of cardiovascular disease, and glucose metabolism status. The results of these studies are confirmed by the above-mentioned data from Hendriks et al. [37], according to which both alcohol use disorder and no alcohol use are among the 15 risk factors for YOD. Therefore, our results obtained for the SIVD patients can definitely reflect the heterogeneity of the sample, in which there were a small number of patients who abused alcohol, but a larger proportion of patients who were characterized by complete abstinence from alcohol; this could be one of the factors that provoked the development of dementia.

As for biochemical indicators (Table 2), when compared to the control group, statistically significant changes in all four groups of patients were detected in the VWF level (a maximum increase of 86% in the SIVD group). The biochemical indicators, the significant changes of which were noted in three groups out of four, were uric acid (SIVD, AIS, and DM with a maximum increase of 40%) and glycerol (AIS, CCCI, and DM with a maximum increase by 2.3-fold, although there were no data on the SIVD group). The indicators, the significant changes of which were noted in two groups out of four, were glucose (in the DM group with an increase of 20.8%, and in the SIVD group with a decrease by 10.4%), albumin (in the CCCI and SIVD groups with a maximum decrease of 11.2%), α1-AGP (in the DM and CCCI groups with a maximum increase of 2.3 fold), and ADAMTS13 (in the AIS and SIVD groups with a maximum decrease of 18%). The biochemical parameters, the significant changes of which were noted in one group out of four, were as follows: total protein (6.5% decrease), HDL (14% decrease), cholesterol (25% decrease), and Fe (2.5-fold decrease) in the SIVD group. In the DM group, these were GGT (1.8-fold increase) and triglycerides (25% increase). In the AIS group, it was NEFA (75% increase). In the CCCI group, there was not a single specific biochemical indicator that changed in a singular manner. At the same time, the levels of urea, lactate, creatinine, AST, ALT, ALP, LDL, and CK-NAC did not show significant differences from the control in any of the four groups of patients, i.e., they were non-specific for these groups. All of them characterize primarily the functional state of the liver, muscles, and kidneys.

The determination of the concentration of markers of EVs showed that there were no significant differences from the control for the level of CD146 in the studied groups (Figure 1a). For the level of CD133, only a tendency to decrease (0.05 ˂ *p* ˂ 0.1) was revealed in the DM group (Figure 1b). For the level of CD31 and CD147, there was an increase of 2- and 15-fold, respectively, only in the AIS group (Figure 1e,f). For the level of CD144, there was a decrease in the DM and CCCI groups by 78% and 61%, respectively (Figure 1c). The level of the CD34 marker was reduced in the CCCI group (by 84%) and to the maximum extent in the DM group (by 88%, i.e., 9-fold, Figure 1d).

Endothelial-derived microparticle (EMP) production can be induced by TNFα, IL-1β, thrombin, C-reactive protein, plasminogen activator-1, high glucose conditions, and hypoxia. EMPs can directly target the vascular endothelium or bind leukocytes and contribute to their activities in inflammatory disorders [41,42]. Rather than mere biomarkers reflecting generalized vascular injury, EMPs and platelet-derived microvesicles (PMVs) have emerged as potent regulators of intercellular communication with significant biological effects in vascular homeostasis and several pathophysiological responses including inflammation and thrombosis [43]. When searching for biomarkers of endothelial origin, one should keep in mind that the most frequently used markers of endothelial progenitor cells (EPCs) expressed on EVs are CD34, VEGFR-2 (CD309), CD45, CD31, CD144, and CD146 [31,44]. However, not all endothelial markers are strictly specific; many are expressed in the membranes of other cells. Therefore, to determine the state of the ECs of blood vessels, it is necessary to evaluate the “endothelial profile” as a set of markers related to the endothelium. Bone marrow-derived lineage-negative (Lin¯) cells form a heterogeneous population containing a variety of cells at different levels of differentiation, including hematopoietic stem cells (HSCs), mesenchymal stem cells (MSCs), and EPCs [45]. MSCs are defined by the expression of CD105, CD73, and CD90 [46]. EMPs are usually evaluated as CD146 + and CD31+ cells, while EPCs can be evaluated as CD34 + and CD133+, CD34+ and VEGFR2+ cells, or CFU-En colony-forming units [47,48]. EPCs are defined as non-hematopoietic cells (CD45-) expressing markers for stemness (CD34+), immaturity (CD133+), and endothelial maturity (KDR+) [48]. In the absence of hematopoietic (CD45 and CD14) and progenitor markers (CD34 and CD133), circulating EMPs can be identified as the cells expressing the endothelial markers CD146, CD144, VWF, and VEGFR2 [49]. Apoptotic EMPs can be registered as CD144 +, CD31 +, and annexinV+ [50], CD105 +, CD144 +, and AnnexinV+ [43], or merely CD144+ [51]. Further, there is a common hemato-endothelial precursor which is CD31 +, CD34 +, and CD144+ [52]. Importantly, CD4+ lymphocytes are necessary for CFU-En formation in vitro [47]. 

Immune reactions provoked by cerebral ischemia play crucial roles in the pathogenesis of brain damage and contribute to the tissue regeneration processes [53]. Previously, it was demonstrated that aging was associated with changes in lymphocyte subsets and HLA-DR upregulation on T-lymphocytes. Measuring early (CD69, 3–5% of total lymphocytes), middle (CD25, 13%), and late (HLA-DR, 8%) T-lymphocyte activation markers on CD3+ lymphocytes across subjects aged 20 to 100 years revealed that the CD3+ and HLA-DR+ subset increased two-fold with age and included CD4+, CD8+, and CD45RA+ phenotypes, as well as CD8 + and CD57+ subset. The CD3 + and CD25+ subset fell with age, and the CD3+ and CD69+ subset showed no age effect [54]. Interestingly, CD69, CD25, and HLA-DR expression were found to be upregulated after chemotherapy and immunotherapy in cancer patients [55]. 

Our analysis of the main lymphocyte subsets (Table 3) revealed no significant deviations between the groups in either the relative or absolute numbers of cells. Moreover, the analysis of the relative and absolute numbers of CD25+ and HLA-DR+ T-cells (Table 4) revealed no significant deviations between the groups either. It should be noted that our data on the relative content of CD25+ and HLA-DR+ T-cells differ significantly from the reference data of 1999 [54], especially when taking into account age trends. A possible explanation may relate mainly to technical advances and antibody affinities, and to a lesser extent it relates to national dietary traditions and natural-ecological factors.

The primary function of HLA-DR is to present peptide antigens, potentially foreign in origin, to the immune system for the purpose of eliciting or suppressing T-helper responses that eventually lead to the production of antibodies against the same peptide antigen [56]. In the instance of an infection, the peptide is bound into a DR molecule and presented to a few T-cell receptors found on T-helper cells. These cells then bind to antigens on the surface of B-cells stimulating B-cell proliferation. Antigen-presenting cells (macrophages, B-cells, and dendritic cells) are the cells in which DRs are typically found. An increased abundance of the DR ‘antigen’ on the cell surface is often in response to stimulation, and, therefore, DR is also a marker for immune stimulation. Serum soluble interleukin-2 receptor (sIL-2R), tumor necrosis factor (TNF-alpha), but not interferon-gamma (IFN-gamma), were associated with CD3+ and HLA-DR+ lymphocytes. TNF-alpha was associated with CD8+ and CD57+ counts, and sIL-2R and IFN-gamma were associated with the CD3+ CD25+/CD3+ CD4+ ratio [54].

The analysis of B-cell subsets (IgD vs. CD38 expression) revealed a 4-fold decrease in the absolute number of Bm2′ cells (IgD+ and CD38++) in the SIVD group (Table 5). As a counterbalance, there was a 2.7-fold increase in the relative number of Bm3 + Bm4 cells in the SIVD patients, whereas a 1.5-fold elevation in the absolute number of these cells was just a tendency. No other groups showed significant changes to B-cell subsets relative to the HD group of patients. However, the relative number of Bm3 + Bm4 cells in the SIVD patients, as well as the absolute number of the same cells in the AIS patients, were significantly increased in relation to the DM patients. 

An analysis of memory B-cell subsets (IgD vs. CD27 expression) revealed significant deviations between control and all other groups only in the SIVD group, and only in one cell type (Table 6). There was a 5.3-fold relative increase and 3-fold absolute increase in numbers (compared to control) of plasmablasts (IgD–CD27++), i.e., the rapidly produced and short-lived effector cells of the early antibody response (in contrast to plasma cells, which are the long-lived mediators of lasting humoral immunity). 

The analysis of the relative (the percentage of total CD3+ and CD4+ cells) and absolute (the number of cells per 1 μL of whole peripheral blood) numbers of the main ‘polarized’ Th-cell subsets revealed an increase in the relative number of Th17-like cells by 39% and 50% (compared to control) in the AIS and DM groups, respectively (Table 7). The absolute number of Th17-like cells was increased by 52% only in the DM group. 

The analysis of the relative (the percentage within total CD3 + CD4+ cells) and absolute (the number of cells per 1 μL of whole peripheral blood) numbers of Th17 cell subsets revealed an increase in the relative and absolute numbers of Th17.1 cells (CXCR3 + CCR4–) in the DM patients (67% and 94% respectively, compared to control patients) (Table 8). In addition, the relative and absolute numbers of DP Th17 cells (CXCR3 + CCR4+) were increased in the AIS (by 41 and 50%) and DM (by 81 and 64%) groups, whereas in the SIVD patients, only a relative number of this subset of cells was increased by 67%, compared to control patients. 

Finally, analysis of the relative (the percentage within total CD3+ and CD4+ cells) and absolute (the number of cells per 1 μL of whole peripheral blood) numbers of follicular Th-cell subsets revealed no significant deviations from control patients and between the groups (Table 9).

## 3. Discussion

Biomarkers are of tremendous importance for the prediction, diagnosis, and observation of the therapeutic success of common complex and multifactorial diseases. However, the common methods of clinical biochemistry are ineffective or have low efficacy for diagnosing senile diseases at early stages [28,57]. The predictive power of the traditional biomarkers used (e.g., plasma metabolites and body parameters) is apparently not sufficient for the reliable monitoring of stage-dependent pathogenesis. Multidimensional and interdependent patterns of genetic, epigenetic, and phenotypic markers will presumably add a novel quality to predictive values, provided they can be followed routinely along the complete individual disease pathway with sufficient precision [35].

The dramatically increased prevalence of metabolic diseases, such as obesity and diabetes mellitus, and their related complications, including endothelial dysfunction and cardiovascular disease, represent one of the leading causes of death worldwide. Endothelial dysfunction is a systemic disorder in which traditional cardiovascular risk factors, such as aging, gender, hypertension, smoking, hyperglycemia, and dyslipidemia, as well as emerging risk determinants, such as fetal factors, gut microbiome alteration, clonal hematopoiesis, air pollution, and sleep disorders, act synergistically to tip the endothelial balance in favor of vasoconstrictive, pro-inflammatory, and pro-thrombotic phenotypes [32,33]. One of the major challenges of a cardiovascular primary prevention approach is the absence of early biomarkers of endothelial dysfunction, which may be useful for identifying at-risk subjects. Biomarkers of vascular damage have been intensively studied in recent years, in the quest for reliable cardiovascular and brain risk assessment tools that are able to facilitate risk stratification and the early detection of vascular impairment [34]. Appropriate use of a single or a cluster of the biomarkers might enable the following in the near future: (a) the prompt identification of targeted and customized treatment strategies; and (b) the follow-up of the treatment efficacy over time in clinical research and/or in clinical practice.

When assessing the total number of specific and generally different control indicators for each group (Table 10), it should be noted that indicators whose changes had the opposite direction of changes in different groups (glucose in the DM and the SIVD groups) should be considered specific to each of these groups. The maximum number of statistically significant socio-demographic and cognitive deviations from the control HD group was noted in the SIVD group, a total of six, of which three were specific (no other groups had specific socio-demographic deviations). Further, in the same group, the maximum number of statistically significant biochemical deviations from the control was noted, a total of nine (there was no data on the level of glycerol in patients in this group), of which five were specific (including a specific deviation in the level of glucose), which was also the maximum amount compared to other groups. Together with immunological indicators (a total of five, of which four were specific to this group), there were 20 significant changes from the control, including 12 specific, in the SIVD group.

In second place, in terms of the level of reliable deviations, was the DM group which had two cognitive indicators (both are non-specific) and seven biochemical indicators, of which three were specific, including an elevated glucose level. Together with two non-specific EV markers and six immunological parameters (of which three were specific to this group), there were 17 significant deviations from the control, including six specific, in the DM group. 

In third place was the AIS group which had two cognitive and five biochemical indicators, of which one was specific. Together with two specific EV markers and three non-specific immunological parameters, there were 12 significant changes from the control, including three specific, in the AIS group.

The minimal number of biochemical parameters different from the control occurred in the CCCI group with a total of four, of which there was not a single one that was specific to this group. Together with three non-specific socio-demographic and cognitive markers, and two non-specific EVs markers, there were nine significant changes from the control, with no one being specific, in the CCCI group.

We also found a radical decrease in iron levels in the SIVD patients, which was one of the most characteristic features of this group. The recently presented Azalea Hypothesis for Alzheimer’s disease asserts that iron becomes sequestered, leading to a functional iron deficiency that contributes to neurodegeneration [58]. Iron sequestration can occur by iron being bound to protein aggregates, such as amyloid-β and tau, iron-rich structures not undergoing recycling (e.g., due to disrupted ferritinophagy and impaired mitophagy), and the diminished delivery of iron from the lysosome to the cytosol. Normally, the lysosome plays an integral role in cellular iron homeostasis by facilitating both the delivery of iron to the cytosol (e.g., after endocytosis of the iron–transferrin–transferrin receptor complex) and the cellular recycling of iron. Reduced iron availability for biochemical reactions causes cells to respond by acquiring additional iron, resulting in an elevation in the total iron level within the affected brain regions. As the amount of unavailable iron increases, the level of available iron decreases until, eventually, it is unable to meet cellular demands, leading to a functional iron deficiency [58]. 

Of the EV markers studied, CD146 turned out to be one of the most non-specific in the sense of deviations from the control group of elderly people, members of which have (as a rule) a plethora of age-related diseases. Some studies have highlighted the significance of CD146 and its soluble form in angiogenesis and inflammation, having been shown to contribute to the pathogenesis of many inflammatory autoimmune diseases, such as systemic sclerosis, DM, rheumatoid arthritis, inflammatory bowel diseases, and multiple sclerosis [59]. It is a cell adhesion molecule expressed on endothelial cells, as well as on other cells such as mesenchymal stem cells and Th17 lymphocytes [60]. CD146 is not merely an adhesion molecule, but also a cellular surface receptor of miscellaneous ligands, including some growth factors and extracellular matrixes [59]. 

Judging by the level of the progenitor markers CD133 and CD34, vasculogenesis and/or vascular repair was impaired to the maximum extent in the DM group. To a slightly lesser extent, impairments were registered in the patients of the CCCI groups, and even less so in the AIS patients (no significant changes in CD34 and CD144 levels), with a complete absence of such in patients with dementia (Table 5). CD144 (also known as VE-cadherin) is the major component of endothelial adherens junctions and is specific to endothelial cells [61]. Adherens junctions have an important role in the control of vascular permeability, being located at cell-to-cell contacts.

Depletion of CD34+ and KDR+ EPCs is an independent predictor of early subclinical atherosclerosis in healthy subjects and may provide additional information beyond the classic risk factors and inflammatory markers [62]. On the other hand, in a study by Rakkar et al., EPC numbers were higher in stroke patients at all-time points studied, reaching significance at baseline and day 30 [48]. Moreover, they came to the conclusion that baseline EPC counts may serve as a diagnostic marker for stroke but fail to distinguish between different stroke subtypes and predict post-stroke outcomes [48]. We revealed no significant deviations in EPC levels in the AIS group, which may be due to the age-related characteristics of our patients, the level of vascular damage, or the time period after the event, but most likely due to a combination of all these reasons. This is supported by a specific increase in the level of CD31 and CD147 in the AIS group, which indicates damage to EC blood vessels after stroke [63,64,65,66]. It is interesting to note here that moderate or severe coronary artery disease (CAD) was associated with increased levels of apoptotic EMPs (CD144+, CD31+, and annexinV+) and reduced EPC colony-forming capacity, increasing the occurrence of endothelial injuries [50]. EMPs that are CD105+, CD144+, and AnnexinV+ have emerged as active contributors to thromboinflammation and vascular damage [43]. CD31 is the most common of the EV markers for diagnosing and predicting the progression of atherosclerosis at various stages and clinical manifestations [67].

The absence of significant changes of the CD34 and CD144 markers in the AIS and SIVD groups, but a decrease in its level in the CCCI and DM groups, may indicate that in the control group of elderly people, there were diseases of the main and peripheral vessels, including coronary arteries, not associated with cerebrovascular accident or diabetes. Therefore, the background levels of CD34 and CD144 in them were increased relative to healthy young and middle-aged people. Acute cerebrovascular accident in patients of the AIS group causes an additional release of CD31 and, characteristically, CD147. 

CD147/basigin, a transmembrane glycoprotein, is a member of the immunoglobulin superfamily. CD147 is overexpressed in malignant melanoma and plays an important role in cell viability, apoptosis, proliferation, invasion, and metastasis, probably by mediating vascular endothelial growth factor (VEGF) production, glycolysis, and multi-drug resistance [68,69]. As for ECs, CD147 is an “early” adhesion receptor, which can be associated with the β2-adrenergic receptor to initiate the signaling cascades induced, for example, by meningococcus in host cells; β-arrestin-mediated signaling pathways are involved in this mechanism [70]. CD147 gained its greatest popularity due to the COVID-19 pandemic, since, in addition to ACE2, both sialic acid (SA) molecules and CD147 proved relevant host receptors for SARS-CoV-2 entry, which explains the viral attack on multiple types of cells, including erythrocytes, ECs, and neural tissue [71]. This can lead to intracellular iron accumulation due to a deregulated hepcidin-ferroportin axis, with concurrent hemoglobin and iron/calcium dysmetabolism and final ferroptosis. 

CD147 induces the production of matrix metalloproteinases (MMPs), which contribute to secondary damage after stroke by disrupting the BBB and facilitating peripheral leukocyte infiltration into the brain [66]. CD147 surface expression increased significantly on infiltrating leukocytes, astrocytes, and ECs after stroke, but not on resident microglia. However, in another study, it was shown that the expression of CD147 and MMP-9 in primary microglia was also up-regulated [72]. The splenic inflammatory response after cerebral ischemia has been implicated in secondary brain injury, and CD147 was shown to be a key mediator of the spleen’s inflammatory activation [73]. In stroke patients, high levels of serum CD147 24 h after stroke predicted poor functional outcome at 12 months. Brain CD147 levels were correlated with MMP-9 and secondary hemorrhage in post-mortem samples from stroke patients [66]. In milder cases of cerebral infarction, CD147 serves as a promoter of atherosclerosis [65]. 

Aging induces a series of immune-related changes, which is called immunosenescence, playing important roles in many age-related diseases, especially neurodegenerative diseases (NDDs), tumors, cardiovascular diseases, autoimmune diseases, and COVID-19. With relatively healthy aging, the adaptive immune system adjusts to age-related changes and protects the body from most pathogens, although, the generation of naive T/B-cells continues to decline. Centenarians have a large number of anti-inflammatory molecules, such as TGF-β1, IL-10, and IL-1 receptor antagonist (IL-1RA), to counterbalance increased inflammatory molecules, such as IL-1β, IL-6, TNF-α, IL-8, CRP, and CXCL9, achieving a dynamic balance between pro-inflammatory and anti-inflammatory levels [74]. The cytotoxic capability of natural killer (NK) cells in centenarians (up to about 55%), which is very similar to the young groups (about 63%), is higher than that in middle-aged groups (about 33%) [75]. There is a view that immune changes concomitant with aging could be seen as an adaptive physiological response. Therefore, it remains an open question as to whether they should be reversed or just controlled, taking into account that such changes may lead to health issues in the long term [76].

Advanced age affects the distribution of T-helper subtypes Th1 and Th2 in a distinct manner that increases Th2 lymphocyte numbers, and associated pro-inflammatory responses appear to play a pivotal role in atherosclerosis and atherothrombosis [77,78]. The innate immune responses dominated by monocytes/macrophages also play a key role in determining the balance between the progression and regression of atherosclerotic disease [79]. NK cells and cytotoxic T-cells play crucial roles in the recognition and clearance of senescent cells and damaged cells [80], but the age-related accumulation of NK cells may contribute to chronic low-grade inflammation, or inflammaging [81]. By accelerating cellular senescence and apoptosis, chronic low-grade inflammation may promote tissue degeneration, plaque instability, and barrier disruption, and constitute a key factor for vascular remodeling [78,82].

Currently, it is believed that peripheral immune cells can impact the progression of NDDs, either after infiltrating the brain or while staying in the periphery [83,84]. The interaction between peripheral immune cells and the brain is an important component of the neuroimmune axis. Unconventional T-cells, which include natural killer T (NKT) cells, mucosal-associated invariant T (MAIT) cells, γδ T-cells, and other poorly defined subsets, are a special group of T-lymphocytes that recognize a wide range of non-polymorphic ligands, and they are the connection between adaptive and innate immunity. SAD patients have lower naive cells, higher memory cells, and a significant telomere shortening of T-cells [85]. 

Neuroinflammation is accepted as a key disease driver caused by innate microglia activation. Self-antigens would initiate autoreactive effector T-cells (Teffs) that drive pro-inflammatory and neurodestructive immunity, leading to cognitive impairments. The treatment of APP/PS1 mice with Aβ reactive Teffs accelerated memory impairment and systemic inflammation, increased amyloid burden, elevated microglia activation, and exacerbated neuroinflammation [86]. Both Th1 and Th17 Aβ-reactive Teffs progressed AD pathology by downregulating anti-inflammatory and immunosuppressive Tregs, as recorded in the periphery and within the central nervous system. Nevertheless, the aberrant disease-associated effector T-cell immune responses can be controlled by Aβ reactive Tregs. Tregs play a neuroprotective role by suppressing microglia and macrophage-mediated inflammation and modulating adaptive immune reactions; Treg immunomodulatory mechanisms are compromised in AD. Ex vivo expanded Tregs, with amplified immunomodulatory function, suppressed neuroinflammation and alleviated AD pathology in vivo [87].

In our study, the identified changes in immunity indicate the discreteness of vascular pathology in four groups of elderly people relative to conditionally healthy controls. This discreteness does not necessarily reflect the stages (sequence) of observed changes in groups, but this cannot be discounted either. As was mentioned, persistent CBF reduction may evoke AIS, vascular cognitive impairment, or even dementia [12,15]; though the immunological profile or basis for these states was not studied before, more so in conjunction with DM. 

In the AIS group, acute changes or compensatory disorders were identified, associated with the non-specific mobilization of T-cell immunity and the almost complete absence of abnormalities specific to this group as compared to others. In the DM group, there was a maximum response of the immune system, mainly T-cell immunity, apparently due to the prolonged deviation in carbohydrate metabolism; associated with this is intestinal dysbiosis and the presence of pro-inflammatory factors [88,89]. In the CCCI group, the reaction was not as pronounced compared to the other groups of patients, which may indicate the initial stages and/or compensatory nature of organic changes. At the same time, in the SIVD group, there were obviously persistent organic changes in blood vessels, which corresponded to the stage of decompensation. The protective function in patients of the SIVD group was apparently assumed by specific factors, the generation of which was carried out mainly by ECs and liver cells to the detriment of anabolic functions, as evidenced by the reduced levels of total protein, albumin, ADAMTS13, glucose, cholesterol, and HDL, but increased VWF and uric acid levels. 

One of the main questions that arises in this case is the primary/leading biochemical disorders or factors that trigger pathological processes associated with vascular pathology and related diseases. We can speculate that skeletal muscle purine degradation could be an underlying driver of uric acid production, with the final step of uric acid production occurring primarily in a non-muscle cell type, ECs being the principal ones [90,91,92]. If this is true, skeletal muscle fiber purine degradation may represent a therapeutic target to reduce serum uric acid and treat numerous pathologies [90].

Aging is among the principal risk factors for many chronic illnesses such as cancer, cardiovascular disorders, and NDDs. Aging results from the progressive dysregulation of several molecular pathways, and mTOR and AMPK signaling have been suggested to play a role in the complex changes of key biological networks involved in cellular senescence [93,94]. Moreover, multiple factors, including poor nutritional balance, drive immunosenescence progression, one of the meaningful aspects of aging. Unsurprisingly, nutraceutical and pharmacological interventions could help maintain an optimal biological response by providing essential bioactive micronutrients required for the development, maintenance, and expression of the immune response at all stages of life [2]. The interplay among immune dysfunction, chronic tissue inflammation, endothelial damage, and innovative therapeutic approaches, highlights the importance of elucidating these complex processes in developing effective interventions to improve the quality of life in elderly people [95]. Nutraceutical and molecular biology represent new insights in this field. In fact, the first could represent a possible treatment in the prevention or delay of vascular aging; the second could offer new possible targets for potential therapeutic interventions [96,97]. Alongside therapeutic options, dietary nutrients together with healthy lifestyles have a crucial role in the endothelium health-promoting effects [32]. One of the most important criteria can be the level of EPCs: elevated numbers of EPCs with CD34+ and CD133+ co-expressions had a dose-dependent association with decreased AD risk. It is therefore tempting to suggest that all natural and therapeutic factors that elevate the level of EPCs can be therapeutic in reducing AD and related disease risks in the presence of cerebrovascular pathology [98]. Further investigation of the relationship between endothelial biomarkers and their dynamic with functional state and age-related diseases is warranted.

## 4. Materials and Methods

### 4.1. Chemicals

PBS (pH 7.4) was purchased from Biolot (St. Petersburg, Russia). Diagnostic kits were produced by Randox Laboratories (Crumlin, UK). All other reagents were from Sigma-Aldrich (Rockville, MD, USA).

### 4.2. Patients and Sample Preparation

This study was conducted in accordance with the Declaration of Helsinki and approved by the Institutional Ethics Committee of the Research Institute of Hygiene, Occupational Pathology and Human Ecology (Approval No. 3, registration date 2 June 2022). Informed consent was obtained from all subjects involved in this study. 

Blood samples were collected, processed, and stored in accordance with international guidelines [99]. Blood was collected from the subjects on an empty stomach from the cubital vein into BD Vacutainer vacuum tubes with anticoagulants (K3EDTA, heparin, and citrate). The plasma was stored at −70 °C until needed. 

### 4.3. Biochemical and Hematological Analysis

Biochemical parameters were determined on a Sapphire 400 analyzer using commercial RANDOX kits. Immunological and hematological studies of whole blood were performed on the day of blood collection. Baseline parameters and histograms of the distribution of erythrocytes, leukocytes, and platelets by volume were determined using a Medonic hematology analyzer (Boule Diagnostics, Spånga, Sweden).

### 4.4. Von Willebrand Factor and ADAMTS13 

To quantify vWF in blood plasma using the standard method, the Technozym vWF:Ag ELISA kit (Technoclone GmbH, Vienna, Austria) was used. The activity of the von Willebrand factor in the plasma of patients was determined in the agglutination reaction of lyophilized platelets with ristocetin using a set of reagents from the Renam company (Moscow, Russia, No. AG-5). The determination of ADAMTS13 activity (VWF cleaving agent) in plasma was carried out on a Synergy 2 plate fluorometer (BioTech, Hudson, MA, USA) using the fluorescent substrate FRETS-VWF73.

### 4.5. Extracellular Vesicles

The study of EVs was carried out in plasma obtained from whole blood collected in vacuum tubes with EDTA. Samples were processed with sequential centrifugations. In the first step, blood samples were centrifuged two times at 1500× *g* for 20 min at +18 °C, followed by centrifugation at 3000× *g* for 20 min at +4 °C. After each centrifugation step, the supernatants were carefully transferred to new conical tubes. The procedure was performed to eliminate residual cells from the samples. The resulting plasma was aliquoted and stored at −80 °C for further use. 

The phenotyping of circulated extracellular vesicles was performed according to the official recommendations from the International Society for Extracellular Vesicles [100,101,102] and previously described and experimentally approved by our group [103,104,105]. 

Immunofluorescence staining of the samples was carried out using the following antibodies (BioLegend, San Diego, CA, USA): anti-CD31 FITC, anti-CD34 PE/Dazzle™ 594, anti-CD133 PE/Cy7, anti-CD144 APC, anti-CD146 PE, and anti-CD147 PE. Sample phenotypes were investigated using 4 panels of antibodies, performed separately in appropriate tubes. A total of 50 µL of each sample was stained with 0.5 µL of each monoclonal antibody for 25 min at 20 °C in the dark. The isotype controls were as follows: Alexa Fluor 488 Mouse IgG1, κ Isotype Ctrl (BioLegend Inc., San Diego, CA, USA, clone: MOPC-21, cat. 400129, concentration 200 µg/mL), PE Mouse IgG1, κ Isotype Ctrl (BioLegend Inc., San Diego, CA, USA, clone: MOPC-21, cat. 400114, concentration 200 µg/mL), and APC Mouse IgG1, κ Isotype Ctrl (BioLegend Inc., San Diego, CA, USA, clone: MOPC-21, cat. 400122, concentration 200 µg/mL). They were stained in the same conditions and concentrations as the appropriate fluorochrome-marked antibodies. 

A total of 20 µL of stained samples were added to 180 µL of phosphate-buffered saline (PBS), resulting in a 10-fold dilution. This 10-fold dilution was then serially diluted 3 times with 100 µL of buffer with reagent added to 900 µL of DPBS. All diluted samples were measured with the three serial dilutions, starting from 10-fold and finishing with 1000-fold, in the series used for calculating EV concentration in the starting material.

A buffer-only control of PBS (Biolot, St. Petersburg, Russia) and a buffer with reagent controls (0.5 µL of each monoclonal antibody in 50 µL of PBS) were recorded at the same flow cytometer acquisition settings as all other samples, including the triggering threshold, voltages, and flow rate. All these controls were diluted 5 times with PBS in the same way as the samples to allow comparisons between serial diluted stained samples. The buffer with reagent control had the same event rate as the buffer-only control.

Unstained controls were measured at the same dilution as stained samples and isotype controls. The flow cytometer acquisition settings for all these controls were unchanged including the triggering threshold, voltages, and flow rate. No substantial changes in scatter or fluorescence signals were observed between the unstained and matched isotype controls.

A working dilution of stained samples to avoid the effect of coincidence was performed. Samples were serially diluted. Dilutions of 1:10, 1:100, and 1:1000 demonstrated a linear correlation between the dilution factor and measured event count, whereas the fluorescence of the events and scatter intensity did not change. Thus, the dilution of 1:10 was used to calculate EV concentration and report the results of immunostaining and phenotyping.

The phenotyping of EVs was performed using a Cytoflex S (Beckman Coulter, Chaska, MN, USA) high-sensitivity flow cytometer equipped with 405 nm (violet), 488 nm (blue), 638 nm (red), and 561 nm (yellow-green) lasers. The detection was triggered on a 405 nm laser at the threshold of 2000 arbitrary units. Instrument calibration setup was performed using the Cytometry Sub-Micron Particle Size Reference Kit, Molecular probes by Life Technologies), and Megamix-Plus FSC and Megamix-Plus SSC (Biocytex, Marseille, France) containing FITC-labeled reference beads of various diameters. The samples were enumerated using the integral flow rate sensors. The sample flow rate was set at 120 µL min^−1^. Stained objects were detected using the side scatter from the violet 405 nm laser and the appropriate use of the fluorochrome-labeled antibodies light pass channel. 

To prove the presence of the membrane in studied objects, additional controls with detergent treatment were performed. Investigated plasma samples were stained with monoclonal fluorescent-labeled antibodies. Then, equal volumes of either PBS or 2% Triton X100 in PBS were incubated with stained samples at room temperature in the dark for 20 min. Samples were phenotyped as described above. The ratio of objects lost was no less than 90%. 

All obtained results of EV phenotyping were analyzed using the CytExpert v.2.4 (Beckman Coulter, Chaska, MN, USA) and Kaluza 2.1 (Beckman Coulter, Chaska, MN, USA) software. The representative results of EV phenotyping are presented in Appendix A.

Statistical analysis was performed using Statistica 7.0 (StatSoft, Oklahoma, OK, USA) and GraphPad Prism 8 (GraphPad Software Inc., San Diego, CA, USA) software. All results were presented as median and interquartile range—Me (25; 75). To determine the significance of differences, the Kruskal–Wallis test and Dunn’s multiple comparisons test were used. Differences were considered significant at *p* < 0.05. 

### 4.6. Immunological Analysis

An analysis was carried out on the main subsets of “polarized” T-helper cells and the main B-cell subsets in peripheral blood in accordance with the recommendations outlined earlier [106,107]. 

#### 4.6.1. Flow Cytometry B-Cell Immunophenotyping

B-cell whole peripheral blood samples (200 μL) were stained with the following anti-human monoclonal antibodies: IgD-Alexa Fluor 488 (cat. 348216, BioLegend, Inc., San Diego, CA, USA), CD38-PE (cat. A07779, Beckman Coulter, Brea, CA, USA), CD5-ECD (cat. A33096, Beckman Coulter, Brea, CA, USA), CD27-PC7 (cat. A54823, Beckman Coulter, Brea, CA, USA), CD19-APC/Cy7 (cat. 302218, BioLegend Inc., San Diego, CA, USA), and CD45-Krome Orange (cat. A96416, Beckman Coulter, Brea, CA, USA). All antibodies were utilized at the dilutions that were recommended by the manufacturers. After incubation at room temperature in the dark for 15 min, red blood cells were lysed for 15 min by adding 2 mL of VersaLyse Lysing Solution (Beckman Coulter, Inc., Brea, CA, USA) supplied with 50 μL IOTest 3 Fixative Solution (Beckman Coulter, Inc., Brea, CA, USA). Next, cells were washed (7 min, 330 g) twice with a buffer (sterile phosphate-buffered saline (PBS) containing 2% of heat-inactivated fetal bovine serum, Sigma-Aldrich, St. Louis, MO, USA) and were resuspended in 0.5 mL of PBS containing 2% of neutral buffered formalin solution (Sigma-Aldrich, St. Louis, MO, USA). Sample acquisition was performed using a Navios flow cytometer (Beckman Coulter, Inc., Brea, CA, USA), equipped with 405, 488, and 638 nm lasers. At least 5000 CD19+ B-cells were collected for analysis from each sample.

#### 4.6.2. T-Cell Immunophenotype by Flow Cytometry

Eight-color flow cytometry was used to analyze the surface phenotype (CD3, CD4, CD45RA, and CD62L) and chemokine receptors (CXCR5, CCR6, CXCR3, and CCR4) on peripheral blood lymphocytes. The antibodies used were as follows: main maturation Th cell subsets phenotyping was performed with CD45RA-FITC (cat. IM0584U, Beckman Coulter, Brea, CA, USA), CD62L-PE (cat. IM2214U, Beckman Coulter, Brea, CA, USA), CD3-APC-AF750 (cat. A94680, Beckman Coulter, Brea, CA, USA), and CD4-PacB (cat. B49197, Beckman Coulter, Brea, CA, USA); chemokine receptor profile was assessed by using CXCR5-PerCP/Cy5.5 (CD185, cat. 356910, BioLegend Inc., San Diego, CA, USA), CCR6-PE/Cy7 (CD196, cat 353418, BioLegend Inc., San Diego, CA, USA), CXCR3-APC (CD183, cat. 353708, BioLegend, Inc., USA), and CCR4-BV510 CD194, cat. 359416, BioLegend Inc., San Diego, CA, USA). Staining protocols were performed in accordance with the manufacturer’s recommendations. In brief, 200 μL of whole peripheral blood sample were stained with the antibody cocktail noted above (all antibodies were utilized at the dilutions that were recommended by the manufacturers) in the dark at room temperature for 15 min, followed by red blood cell lysis with 2 mL of VersaLyse Lysing Solution (Beckman Coulter, Brea, CA, USA) with 50 μL IOTest 3 Fixative Solution (Beckman Coulter, Brea, CA, USA; incubation time of 15 min in the dark at room temperature). Next, all samples were washed once with PBS and centrifuged for 7 min at 330× *g*, resuspended in 500 μL of PBS with 2% of neutral formalin (cat. HT5011-1CS, Sigma-Aldrich, Rockville, MD, USA). Finally, they were analyzed by flow cytometry with a Navios flow cytometer (Beckman Coulter, Brea, CA, USA). At least 40,000 CD3+ and CD4+ Th cells were collected for each sample. The data collected were analyzed with Kaluza software v2.1 (Beckman Coulter, Brea, CA, USA).

### 4.7. Statistical Analysis

The flow cytometry data were analyzed with Kaluza 2.1 software (Beckman Coulter, Inc., Brea, CA, USA). All the statistical analysis of data was carried out with STATISTICA Version 8.0 (StatSoft Inc., Tulsa, OK, USA) and GraphPad Prism Version 5.0 (USA). The obtained data were tested for normality of distribution using the Shapiro–Wilk test (the HC group contained less than 50 patients). The quantitative data are presented as median and quartile ranges (Med (Q25; Q75)). The statistical analysis was performed with the Kruskal–Wallis test and Dunn’s multiple comparisons test. Only significant differences from the control group are shown (p1), according to Dunn’s multiple comparisons test. The differences between the groups were considered significant when *p* values were <0.05.

## 5. Conclusions

Taking into account all four groups of indicators (socio-demographic, lifestyle and cognitive data, biochemical and immunological indicators, and CD-markers of the EVs), the largest number of deviations from control was registered in patients with dementia of vascular origin (SIVD group), a total of 20, of which 12 were specific. However, judging by the number of changes in the immunological parameters and the endothelial CD-markers of EVs, vascular repair was impaired to the maximum extent in the DM group. To a slightly lesser extent, impairments were registered in patients from the CCCI groups, and even less so in AIS patients, with a complete absence of such in patients with dementia (SIVD group). Although microvascular damage seemed to be maximal in the SIVD group when considering the biochemical indicators VWF and ADAMTS13. In the DM group, there was a maximum response of the immune system, mainly through Th17-like cells. In the CCCI group, the reaction was not as pronounced as in other groups of patients, which may indicate the initial stages and/or compensatory nature of organic changes (remodeling). At the same time, in the SIVD group, there was obviously persistent remodeling in blood vessels, which corresponded to the stage of decompensation, chronic inflammation, and a decrease in the anabolic function of the liver and other tissues.

## Figures and Tables

**Figure 1 ijms-25-01888-f001:**
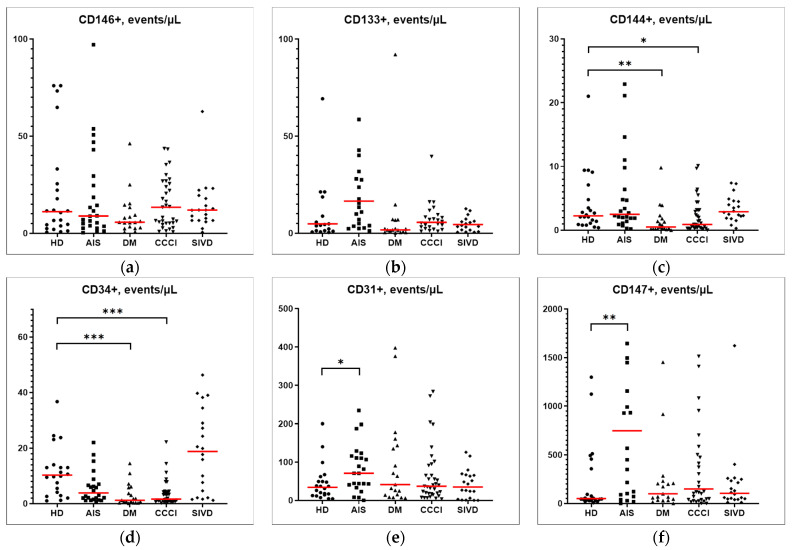
CD markers of endothelial EVs. (**a**) the level of CD146+ marker; (**b**) the level of CD133+ marker; (**c**) the level of CD144+ marker; (**d**) the level of CD34 marker; (**e**) the level of CD31+ marker; (**f**) the level of CD147+ marker. The quantitative data are presented as median (red line) and quartile ranges (Med (Q25; Q75). The statistical analysis was performed with the Kruskal–Wallis test and Dunn’s multiple comparisons test; only significant differences from the control group are shown, according to Dunn’s multiple comparisons test. * *p* ˂ 0.05; ** *p* ˂ 0.01 *** *p* ˂ 0.001.

**Table 1 ijms-25-01888-t001:** The patient cohorts, their socio-demographic and lifestyle scores, and cognitive status.

	HD	AIS	DM	CCCI	SIVD
Total patients (120)	23	24	21	32	20
Age Me (min, max)	65 (60, 72)	67.5 (60, 75)	67 (60, 73)	68.5 (60, 77)	66 (60, 74)
Men (52)	11	10	9	14	8
Age of men Me (min, max)	64 (60, 70)	66 (60, 72)	66 (60, 71)	67 (60, 74)	65 (60, 70)
Women (68)	12	14	12	18	12
Age of women Me (min, max)	66 (61, 72)	69 (62, 75)	68 (62, 73)	69 (63, 77)	67 (61, 74)
BMI Me (min, max)	21.4 (19.1, 27.9)	22.9 (20.2, 29.1)	25.8 (21.9, 33.8)	23.3 (20.3, 29.9)	21.8 (19.4, 32.9)
Smoking	1 (1; 8)	4 (1; 9)	2.5 (1; 9)	1 (1; 9)	1 (1; 7)
					p2 = 0.019
Alcohol	3 (1; 7)	3 (1; 4)	3 (1; 5)	3 (1; 5)	1 (1; 9)
					p1 = 0.0021
Socialization	2 (1; 5)	2 (1; 4)	3 (1; 5)	2 (1; 5)	1 (1; 2)
					p1 = 9.6 × 10^−6^
					p2 = 2.5 × 10^−6^
					p3 = 2.0 × 10^−6^
					p4 = 2.2 × 10^−5^
Life satisfaction	4 (1; 4)	3 (2; 4)	4 (1; 4)	3 (2; 4)	3 (1; 3)
				p1 = 0.0022	p1 = 0.0002
Physical activity	3 (1; 4)	3 (1; 4)	2 (2; 3)	2 (1; 5)	2 (1; 2)
					p1 = 0.0018
					p2 = 0.0007
Night sleep	3 (1; 3)	1.5 (1; 3)	3 (1; 3)	2 (1; 3)	3 (1; 3)
MOCA	27.0 (25.0; 28.0)	23.0 (17.0; 29.0)	23.0 (16.0; 28.0)	24.0 (15.0; 29.0)	12.0 (10.0; 19.0)
					(n = 20)
					p1 = 1.1 × 10^−16^
		p1 = 3.9 × 10^−6^	p1 = 3.9 × 10^−4^	p1 = 1.7 × 10^−4^	p2 = 1.2 × 10^−4^
					p3 = 2.7 × 10^−6^
					p4 = 1.2 × 10^−7^
MMSE	29.0 (28.0; 30.0)	26.0 (23.0; 30.0)	27.0 (23.0; 30.0)	28.0 (24.0; 30.0)	14.0 (12.0; 20.0)
					(n = 20)
					p1 = 9.23 × 10^−16^
		p1 = 8.18 × 10^−6^	p1 = 0.0028	p1 = 0.0047	p2 = 0.000188
					p3 = 6.37 × 10^−7^
					p4 = 3.50 × 10^−9^

Notes: The quantitative data are presented as median and quartile ranges (Med (Q25; Q75). The statistical analysis was performed with the Kruskal–Wallis test and Dunn’s multiple comparisons test; p1, p2, p3, p4, and p5—the differences with HD, AIS, DM, CCCI, and SIVD, respectively—were significant according to the Dunn’s multiple comparisons test. **Individual smoking scores** were ranked as follows: 1—never smoked; 2—up to 5 cigarettes/day, up to 10 years; 3—up to 10 cigarettes/day, 10+ years; 4—10–20 cigarettes/day, up to 10 years; 5—10–20 cigarettes/day, 10–20 years; 6—20+ cigarettes/day, up to 10 years; 7—20+ cigarettes/day 10–20 years; 8—10–20 cigarettes/day 20+ years; 9—20+ cigarettes/day, 20+ years. **Individual alcohol scores** were ranked as follows: 1—never used; 2—low-alcohol drinks (beer/wine) occasionally; 3—strong alcoholic drinks (vodka, cognac) occasionally; 4—low-alcohol drinks weekly; 5—strong alcoholic drinks weekly; 6—low-alcohol drinks several times a week; 7—strong alcoholic drinks several times a week; 8—low-alcohol drinks daily; 9—strong alcoholic drinks daily. **Individual socialization scores** were ranked as follows: 1—lives alone; +1 point for spouse; +1 point for children; +1 point for grandchildren; + 1 point for other relatives; +1 point for noting good family relationships. **Individual life satisfaction scores** were ranked as follows: 1—not satisfied; 2—more or less satisfied; 3—completely satisfied; 4—completely satisfied and happy. **Individual physical activity scores** were ranked as follows: 1—physical activity is practically absent (bedridden patient); +1 point for each type of activity (mostly sedentary lifestyle; working in the garden; daily walks; daily gymnastics; Nordic walking; visiting the pool). **Individual night sleep scores** were ranked as follows: 1—often insomnia, difficulty falling asleep, nightmares; 2—whatever happens, anything can happen; 3—normal sleep.

**Table 2 ijms-25-01888-t002:** Diagnostic indicators of the biochemical profile of elderly people (60–78 years old) with cerebrovascular accidents (AIS and CCCI), diabetes mellitus, and those suffering from SIVD. Medians are indicated (min; max).

Indicator, Ref. BCA Range	HD	AIS	DM	CCCI	SIVD
Glu, 3.9–5.8 mmol/L	4.8 (4.3; 5.9)	5.0 (4.4; 6.6)	5.8 (4.4; 7.8)	5.0 (3.9; 6.3)	4.3 (3.5–9.7)
	(*n* = 23)	(*n* = 23)	(*n* = 20)	(*n* = 30)	(*n* = 20)
			p1 = 0.0019		p1 = 0.0035
Total Protein, 66–87 g/L	69.7 (63.5; 80.9)	72.4 (65.3; 74.1)	70.5 (65.7; 80.3)	68.6 (61.1; 77.5)	65.2 (47.7; 73.9)
	(*n* = 23)	(*n* = 24)	(*n* = 20)	(*n* = 32)	(*n* = 20)
					p1 = 0.0021
ALB, 38–44 g/L	41.8 (37.9; 44.6)	40.5 (36.2; 45.7)	41.3 (25.7; 45.6)	40.8 (37.1; 43.9)	37.1 (25.7; 42.5)
	(*n* = 23)	(*n* = 24)	(*n* = 20)	(*n* = 32)	(*n* = 20)
				p1 = 0.026	p1 = 3.0 × 10^−8^
AST 0-40 U/L	26.0 (16.0; 40.0)	28.5 (20.0; 58.0)	29.0 (19.0; 145.0)	29.0 (20.0; 61.0)	25.0 (14.0; 74.0)
	(*n* = 23)	(*n* = 24)	(*n* = 20)	(*n* = 32)	(*n* = 20)
ALT, 0-40 U/L	23.0	24.3	24.8	25.0	21.4
	(11.0; 56.9)	(12.1; 57.2)	(12.5; 127.2)	(6.0; 69.5)	(10.1; 108.9)
	(*n* = 23)	(*n* = 24)	(*n* = 20)	(*n* = 32)	(*n* = 20)
GGT, 11–50 U/L	22.0 (6.0; 147.8)	33.5 (1.0; 219.0)	40.5 (13.0; 466.0)	32.0 (1.0; 320.0)	27.0 (11.0; 403.0)
	(*n* = 23)	(*n* = 24)	(*n* = 20)	(*n* = 32)	(*n* = 20)
			p1 = 0.0038		
ALP, 30–120 U/L	24.3 (20.8; 111.0)	22.6 (19.9; 114.0)	37.7 (20.9; 116.0)	23.0 (20.4; 168.0)	20.4 (14.1; 23.4)
	(*n* = 23)	(*n* = 24)	(*n* = 20)	(*n* = 32)	(*n* = 20)
Urea, 2.5–7.5 mmol/L	5.4 (2.5; 8.5)	5.7 (2.3; 9.6)	4.8 (2.4; 11.9)	5.9 (3.0; 11.9)	6.2 (3.0; 17.9)
	(*n* = 23)	(*n* = 24)	(*n* = 20)	(*n* = 32)	(*n* = 20)
CREA 2R, 53–110 umol/L	79 (59; 138)	83 (61; 131)	85 (61; 104)	78.0 (56; 241)	85 (68; 112)
	(*n* = 23)	(*n* = 24)	(*n* = 20)	(*n* = 32)	(*n* = 20)
Uric acid, 202–416 umol/L	250.0	330.0	349.5	294.5	330.5
	(118.0; 393.0)	(201.0; 492.0)	(93.0; 476.0)	(85.0; 510.0)	(97.0; 431.0)
	(*n* = 23)	(*n* = 24)	(*n* = 20)	(*n* = 32)	(*n* = 20)
		p1 = 0.0039	p1 = 0.0037		p1 = 0.0282
Chol, 3.2–5.6 mmol/L	5.9 (4.2; 8.1)	5.4 (2.7; 7.6)	5.2 (3.1; 7.4)	6.2 (3.3; 8.7)	4.4 (2.6; 6.1)
	(*n* = 23)	(*n* = 24)	(*n* = 20)	(*n* = 32)	(*n* = 20)
					p1 = 3.0 × 10^−5^
TRIGS, 0.1–2.29 mmol/L	1.2 (0.5; 2.4)	1.2 (0.5; 5.6)	1.5 (0.1; 3.7)	1.3 (0.4; 5.0)	1.1 (0.7; 4.7)
	(*n* = 23)	(*n* = 24)	(*n* = 20)	(*n* = 32)	(*n* = 20)
			p1 = 0.0489		
NEFA, 0.71–2.07 0.1–0.9 mmol/L	0.4 (0.1; 0.9)	0.7 (0.1; 1.2)	0.4 (0.1; 1.1)	0.4 (0.1; 1.3)	0.3 (0.1; 0.7)
	(*n* = 23)	(*n* = 24)	(*n* = 20)	(*n* = 32)	(*n* = 20)
		p1 = 0.001			
Glyc, 10–50 mmol/L	40.3	74.2	93.4	75.8	---
	(13.1; 133.1)	(19.7; 185.2)	(40.8; 169.6)	(30.0; 335.9)	
	(*n* = 14)	(*n* = 21)	(*n* = 20)	(*n* = 32)	
		p1 = 0.0050	p1 = 0.0003	p1 = 0.0004	
LAC, 0.5–2.22 mmol/L	2.2 (0.7; 1.2)	2.0 (1.2; 5.0)	2.2 (0.8; 4.8)	1.7 (0.2; 4.8)	1.9 (0.4; 3.0)
	(*n* = 23)	(*n* = 24)	(*n* = 20)	(*n* = 32)	(*n* = 20)
HDL R, 1.04–1.55 mmol/L	1.4 (0.8; 2.6)	1.3 (0.8; 1.9)	1.3 (0.9; 1.8)	1.5 (0.8; 2.9)	1.2 (0.7; 1.5)
	(*n* = 19)	(*n* = 24)	(*n* = 20)	(*n* = 32)	(*n* = 20)
					p1 = 0.0113
LDL R, 1.7–3.5 mmol/L	3.9 (2.6; 6.5)	4.0 (1.6; 6.8)	3.7 (1.6; 5.2)	4.0 (1.9; 6.2)	2.7 (1.3; 3.7)
	(*n* = 23)	(*n* = 24)	(*n* = 20)	(*n* = 32)	(*n* = 20)
a1-AGP, 0.5–1.2 g/L	0.7 (0.1; 3.1)	1.0 (0.1; 2.4)	1.5 (0.2; 3.1)	1.6 (0.1; 4.6)	0.9 (0.5; 3.4)
	(*n* = 23)	(*n* = 24)	(*n* = 20)	(*n* = 32)	(*n* = 20)
			p1 = 0.0221	p1 = 0.0029	
CK-NAC, 24–195 U/L	149.0	172.5	145	124.5	72.5
	(27.0; 522.0)	(28.0; 290.0)	(70.0; 247.0)	(41.0; 293.0)	(24.0; 261.0)
	(*n* = 23)	(*n* = 24)	(*n* = 20)	(*n* = 32)	(*n* = 20)
Fe, 10.6–28.3 umol/L	27.1	17.9	23.1	22.4	11.0
	(6.4; 82.3)	(3.0; 71.5)	(4.1; 57.1)	(4.9; 71.5)	(7.2; 18.3)
	(*n* = 23)	(*n* = 24)	(*n* = 20)	(*n* = 32)	(*n* = 20)
					p1 = 4.0 × 10^−6^
VWF, mU/mL	0.7 (0.2; 1.4)	1.1 (0.7; 4.5)	1.0 (0.2; 2.7)	1.2 (0.6; 4.4)	1.3 (0.5; 3.9)
	(*n* = 23)	(*n* = 24)	(*n* = 20)	(*n* = 32)	(*n* = 20)
		p1 = 0.0021	p1 = 0.0043	p1 = 0.0003	p1 = 5.0 × 10^−5^
ADAMTS13, % activity	1.1 (0.6; 1.1)	1.0 (0.8; 1.3)	1.0 (0.7; 1.3)	1.0 (0.6; 1.2)	0.9 (0.2; 1.1)
	(*n* = 23)	(*n* = 17)	(*n* = 19)	(*n* = 31)	(*n* = 19)
		p1 = 0.0162			p1 = 0.0002

Notes: AST—aspartate aminotransferase, GGT—gamma-glutamyltransferase, FFA—free (non-esterified) fatty acids, VWF—von Willebrand factor, ALP—alkaline phosphatase, BCA—biochemical analyzer. The quantitative data are presented as median and quartile ranges (Med (Q25; Q75). The statistical analysis was performed with the Kruskal–Wallis test and Dunn’s multiple comparisons test; only significant differences from the control group are shown (p1), according to Dunn’s multiple comparisons test.

**Table 3 ijms-25-01888-t003:** The relative (%, the percentage within the total lymphocyte subset) and absolute (#, the number of cells per 1 μL of whole peripheral blood) numbers of the main lymphocyte subsets in the patient groups.

Lymphocytes Subsets		HD (*n* = 17)	AIS (*n* = 24)	DM (*n* = 20)	CCCI (*n* = 33)	SIVD (*n* = 14)
T-cells	%	74.01 (69.47; 81.66)	75.81 (69.85; 78.87)	75.32 (70.85; 78.54)	73.93 (68.44; 77.71)	82.34 (71.18; 83.68)
(CD3+)	#	1797 (1262; 1873)	1680 (1353; 2415)	1590 (1368; 2239)	1560 (1100; 1901)	1619 (1255; 2188)
CD4+ T-cells	%	48.34 (43,55; 50,28)	49.51 (39.09; 52.71)	51.42 (45.03; 56.38)	47.61 (42.45; 56.36)	46.71 (44.36; 51.31)
(CD3 + CD4+)	#	994 (855; 1290)	1082 (898; 1306)	1060 (874; 1348)	953 (713; 1455)	834 (689; 1391)
CD8+ T-cells	%	21.97 (17.76; 28.84)	23.90 (18.56; 30.22)	22.26 (19.18; 24.85)	22.18 (16.28; 30.57)	27.30 (18.57; 35.87)
(CD3 + CD8+)	#	507 (374; 641)	571 (408; 771)	509 (325; 709)	411 (308; 657)	524 (326; 742)
CD4/CD8 ratio		2.29 (1.40; 3.11)	1.82 (1.46; 2.71)	2.41 (1.65; 2.70)	1.98 (1.58; 3.19)	1.86 (1.29; 2.54)
Tregs cells	%	3.24 (2.59; 3.60)	2.72 (2.43; 3.39)	2.79 (2.19; 3.62)	2.53 (2.17; 3.20)	2.67 (1.70; 3.86)
(CD3 + CD4 + CD25hi)	#	66 (60; 86)	68 (53; 92)	62 (45; 90)	51 (38; 73)	49 (33; 58)
NK cells	%	11.31 (8.61; 16.47)	13.64 (10.10; 19.05)	12.79 (9.32; 16.39)	13.86 (7.30; 17.98)	9.50 (6.40; 13.70)
(CD3-CD56+)	#	259 (212; 320)	298 (217; 437)	239 (196; 398)	271 (163; 374)	183 (104; 363)
B-cells	%	10.37 (8.03; 12.66)	9.89 (6.92; 11.07)	10.03 (5.56; 12.57)	9.02 (6.59; 11.82)	7.88 (4.77; 12.22)
(CD19+)	#	240 (150; 327)	205 (137; 299)	187 (124; 264)	173 (91; 328)	151 (100; 232)

Note: The quantitative data (relative and absolute numbers of cells) are presented as median and quartile ranges (Med (Q25; Q75). The statistical analysis was performed with the Kruskal–Wallis test and Dunn’s multiple comparisons test; only significant differences are shown; p1, p2, p3, p4, and p5—the differences with HD, AIS, DM, CCCI, and SIVD, respectively—were significant according to Dunn’s multiple comparisons test.

**Table 4 ijms-25-01888-t004:** The relative (%, the percentage within the T-cell subset) and absolute (#, the number of cells per 1 μL of whole peripheral blood) numbers of CD25+ and HLA-DR+ T-cells in the patient groups.

T-Cell Subsets		HD (*n* = 17)	AIS (*n* = 24)	DM (*n* = 20)	CCCI (*n* = 33)	SIVD (*n* = 14)
CD25 expression by different T-cell subsets:						
CD25+ T-cells	%	17.08 (13.94; 22.21)	19.88 (17.32; 24.73)	22.98 (17.72; 31.10)	18.72 (13.03; 24.32)	23.71 (16.77; 25.20)
	#	278 (212; 414)	322 (274; 513)	367 (253; 474)	252 (178; 387)	314 (258; 522)
CD25+ CD8+ T-cells	%	3.98 (2.40; 8.20)	5.50 (3.20; 10.38)	5.24 (2.89; 8.12)	4.39 (2.34; 7.85)	5.16 (3.69; 6.02)
	#	19 (15; 32)	25 (18; 66)	25 (16; 38)	18 (11; 32)	31 (17; 36)
CD25+ CD4+ T-cells	%	35.70 (30.41; 41.32)	39.72 (33.77; 49.73)	43.99 (33.34; 55.17)	35.12 (25.36; 45.91)	45.25 (36.75; 50.34)
	#	355 (274; 490)	425 (342; 584)	467 (300; 609)	322 (230; 459)	349 (295; 717)
HLA-DR expression by different T-cell subsets:						
HLA-DR+ T-cells	%	4.51 (3.32; 6.58)	5.85 (3.93; 10.03)	7.32 (4.07; 10.00)	5.61 (4.40; 8.56)	6.95 (3.87; 18.74)
	#	74 (59; 107)	99 (64; 189)	132 (59; 161)	91 (54; 143)	86 (58; 353)
HLA-DR+ CD8+ T-cells	%	10.41 (8.30; 15.79)	13.01 (8.84; 19.94)	18.78 (9.25; 24.39)	15.11 (9.93; 21.19)	17.58 (10.44; 37.73)
	#	53 (33; 77)	78 (48; 156)	101 (30; 144)	58 (35; 106)	79 (39; 343)
HLA-DR+ CD4+ T-cells	%	3.46 (2.40; 5.34)	4.58 (2.84; 8.42)	5.96 (3.62; 8.08)	5.05 (3.73; 6.86)	4.41 (2.45; 12.18)
	#	38 (28; 53)	55 (30; 88)	61 (35; 81)	53 (33; 77)	37 (24; 90)
HLA-DR+ Tregs cells	%	15.00 (9.40; 18.20)	16.10 (11.62; 23.83)	18.98 (15.81; 24.13)	17.07 (12.57; 22.43)	17.74 (11.32; 19.98)
	#	10 (7; 11)	11 (7; 15)	11 (8; 18)	9 (6; 13)	9 (5; 14)

Note: The quantitative data (relative and absolute numbers of cells) are presented as median and quartile ranges (Med (Q25; Q75). The statistical analysis was performed with the Kruskal–Wallis test and Dunn’s multiple comparisons test; only significant differences are shown; p1, p2, p3, p4, p5—the differences with HD, AIS, DM, CCCI, and SIVD, respectively—were significant according to Dunn’s multiple comparisons test.

**Table 5 ijms-25-01888-t005:** The relative (%, the percentage within total lymphocyte subset) and absolute (#, the number of cells per 1 μL of whole peripheral blood) numbers of B-cell subsets (IgD vs. CD38 expression) in the patient groups.

Lymphocytes Subsets		HD (*n* = 17)	AIS (*n* = 24)	DM (*n* = 20)	CCCI (*n* = 33)	SIVD (*n* = 14)
Bm1, IgD + CD38–	%	19.05 (9.52; 23.71)	12.27 (7.75; 16.31)	12.42 (7.93; 19.24)	12.37 (9.85; 19.88)	11.83 (8.21; 13.51)
	#	34 (21; 74)	25 (17; 40)	27 (17; 48)	29 (18; 50)	20 (11; 25)
Bm2, IgD + CD38+	%	57.27 (49.59; 67.35)	55.87 (39.93; 63.16)	58.82 (49.60; 65.69)	56.98 (48.15; 65.56)	55.86 (46.64; 63.96)
	#	139 (82; 193)	126 (65; 184)	131 (72; 187)	120 (68; 210)	94 (41; 152)
Bm2′, IgD + CD38++	%	7.82 (3.79; 9.27)	7.09 (2.96; 12.35)	6.80 (3.66; 10.41)	8.08 (5.09; 10.67)	3.11 (0.96; 6.85)
	#	20 (13; 28)	19 (5; 40)	14 (9; 21)	14 (8; 33)	5 (1; 12)
						p1 = 0.0329
Bm3 + Bm4, IgD–CD38++	%	0.73 (0.46; 1.15)	1.38 (0.71; 2.31)	0.84 (0.40; 1.26)	0.94 (0.74; 1.43)	1.94 (1.10; 2.94)
						p1 = 0.0388 p3 = 0.0302
	#	2 (1; 3)	4 (2; 5)	2 (1; 2)	2 (1; 4)	3 (1; 5)
			p3 = 0.0212			
eBm5, IgD–CD38+	%	8.74 (7.60; 11.09)	10.14 (6.97; 14.15)	8.38 (6.70; 12.72)	9.18 (6.47; 12.01)	12.40 (10.22; 14.97)
	#	24 (15; 30)	21 (13; 38)	17 (13; 29)	20 (10; 38)	18 (9; 40)
Bm5, IgD–CD38–	%	7.31 (5.47; 9.37)	9.44 (5.52; 13.91)	9.74 (5.43; 16.15)	9.36 (7.24; 11.26)	11.20 (6.32; 18.75)
	#	20 (12; 23)	22 (12; 38)	18 (15; 24)	21 (12; 32)	14 (9; 32)

Note: The quantitative data (relative and absolute numbers of cells) are presented as median and quartile ranges (Med (Q25; Q75). The statistical analysis was performed with the Kruskal–Wallis test and Dunn’s multiple comparisons test; only significant differences are shown; p1, p2, p3, p4, p5—the differences with HD, AIS, DM, CCCI, and SIVD, respectively—were significant according to Dunn’s multiple comparisons test.

**Table 6 ijms-25-01888-t006:** The relative (%, the percentage within total lymphocyte subset) and absolute (#, the number of cells per 1 μL of whole peripheral blood) numbers of memory B-cell subsets (IgD vs. CD27 expression) in the patient groups.

Lymphocytes Subsets		HD (*n* = 17)	AIS (*n* = 24)	DM (*n* = 20)	CCCI (*n* = 33)	SIVD (*n* = 14)
Naïve B-cells, IgD + CD27–	%	68.45 (57.37; 75.76)	62.64 (45.83; 73.25)	64.69 (60.82; 73.50)	67.56 (59.56; 75.34)	65.85 (51.51; 72.24)
	#	168 (106; 224)	141 (76; 224)	121 (84; 218)	139 (81; 252)	104 (40; 202)
Unswitched memory, IgD + CD27+	%	16.59 (7.62; 21.52)	11.38 (8.02; 16.33)	11.98 (7.67; 18.05)	11.16 (7.59; 14.73)	7.86 (5.92; 12.17)
	#	38 (15; 54)	21 (17; 36)	22 (18; 42)	25 (17; 41)	14 (10; 23)
Switched memory, IgD–CD27–	%	11.64 (10.27; 15.57)	15.80 (10.00; 24.34)	14.71 (12.66; 20.39)	14.89 (11.58; 20.58)	17.36 (13.90; 21.32)
	#	32 (22; 44)	37 (21; 60)	29 (23; 49)	37 (17; 54)	25 (12; 61)
Double-negative, IgD–CD27–	%	3.95 (2.52; 6.24)	4.62 (3.08; 6.41)	3.68 (2.36; 5.00)	3.33 (2.63; 5.12)	5.72 (4.16; 7.95)
	#	10 (7; 13)	9 (6; 16)	8 (5; 11)	7 (5; 12)	8 (5; 14)
Plasmablasts, IgD–CD27++	%	0.23 (0.14; 0.46)	0.31 (0.19; 0.82)	0.28 (0.19; 1.21)	0.35 (0.20; 0.71)	1.22 (1.08; 2.45)
						p1 = 0.0004 p2 = 0.0153 p3 = 0.0290 p4 = 0.0062
	#	1 (0; 1)	1 (0; 2)	1 (0; 2)	1 (0; 2)	3 (2; 4)
						p1 = 0.0063 p2 = 0.0480 p3 = 0.057 p4 = 0.0305

Note: The quantitative data (relative and absolute numbers of cells) are presented as median and quartile ranges (Med (Q25; Q75). The statistical analysis was performed with the Kruskal–Wallis test and Dunn’s multiple comparisons test; only significant differences are shown; p1, p2, p3, p4, p5—the differences with HD, AIS, DM, CCCI, and SIVD, respectively—were significant according to Dunn’s multiple comparisons test.

**Table 7 ijms-25-01888-t007:** The relative (%, the percentage within total CD3+ and CD4+ cells) and absolute (#, the number of cells per 1 μL of whole peripheral blood) numbers of main ‘polarized’ Th-cell subsets in the patient groups.

Main ‘Polarized’ Th-Cell Subsets		HD (*n* = 17)	AIS (*n* = 24)	DM (*n* = 20)	CCCI (*n* = 33)	SIVD (*n* = 14)
Th1-like cells	%	11.42 (10.86; 16.58)	15.47 (11.00; 22.72)	14.06 (9.90; 18.58)	15.75 (12.63; 25.15)	15.63 (10.28; 21.11)
	#	144 (100; 174)	169 (130; 279)	159 (98; 178)	187 (113; 266)	149 (84; 199)
Th2-like cells	%	7.01 (5.22; 7.80)	7.05 (5.66; 8.66)	6.25 (4.84; 9.61)	5.69 (4.38; 7.05)	6.48 (4.67; 6.94)
	#	74 (56; 91)	73 (53; 111)	79 (54; 95)	49 (40; 95)	54 (35; 106)
Th17-like cells	%	18.07 (13.39; 22.96)	25.07 (19.89; 32.93) p1 = 0.0390	27.01 (22.14; 37.75) p1 = 0.0031	20.10 (17.15; 25.85)	23.78 (19.96; 35.64)
	#	186 (153; 240)	261 (190; 347)	282 (212; 391)	217 (151; 322)	210 (171; 305)
				p1 = 0.0104		
Th-like cells	%	9.65 (7.18; 11.90)	10.53 (8.40; 13.68)	10.39 (7.33; 15.01)	8.56 (6.46; 11.88)	11.72 (9.12; 13.33)
	#	92 (79; 122)	106 (77; 155)	110 (81; 138)	85 (48; 130)	102 (69; 148)

Note: The quantitative data (relative and absolute numbers of cells) are presented as median and quartile ranges (Med (Q25; Q75). The statistical analysis was performed with the Kruskal–Wallis test and Dunn’s multiple comparisons test; only significant differences are shown; p1, p2, p3, p4, p5—the differences with HD, AIS, DM, CCCI, and SIVD, respectively—were significant according to Dunn’s multiple comparisons test.

**Table 8 ijms-25-01888-t008:** The relative (%, the percentage within total CD3+ and CD4+ cells) and absolute (#, the number of cells per 1 μL of whole peripheral blood) numbers of Th17-cell subsets in the patient groups.

Th17-Cell Subsets		HD (*n* = 17)	AIS (*n* = 24)	DM (*n* = 20)	CCCI (*n* = 33)	SIVD (*n* = 14)
DN Th17, CXCR3–CCR4–	%	1.85 (1.36; 3.25)	2.29 (1.75; 2.71)	2.57 (1.65; 2.75)	1.98 (1.39; 3.08)	1.56 (1.08; 2.12)
	#	22 (13; 35)	24 (17; 32)	26 (15; 40)	20 (13; 30)	16 (10; 20)
Classical Th17, CXCR3–CCR4+	%	6.41 (4.63; 8.07)	8.58 (7.28; 10.02)	8.17 (6.54; 10.61)	6.49 (5.03; 8.33) p2 = 0.0335	8.10 (5.76; 9.98)
	#	64 (52; 79)	91 (55; 117)	98 (58; 131)	62 (45; 103)	72 (50; 109)
Th17.1, CXCR3 + CCR4–	%	5.46 (3.63; 7.28)	7.81 (5.85; 12.19)	9.13 (7.15; 15.21) p1 = 0.0077	7.01 (4.67; 8.79)	6.85 (5.41; 15.19)
	#	48 (44; 84)	85 (64; 110)	93 (66; 148) p1 = 0.0245	74 (42; 140)	73 (45; 117)
DP Th17, CXCR3 + CCR4+	%	4.17 (3.35; 4.88)	5.88 (5.02; 8.13) p1 = 0.0165	7.53 (5.07; 9.02) p1 = 0.0008	4.96 (4.07; 7.01)	6.97 (5.79; 8.18) p1 = 0.008
	#	42 (34; 45)	63 (42; 92) p1 = 0.0253	69 (54; 85) p1 = 0.0033	51 (35; 67)	56 (41; 88)

Note: The quantitative data (relative and absolute numbers of cells) are presented as median and quartile ranges (Med (Q25; Q75). The statistical analysis was performed with the Kruskal–Wallis test and Dunn’s multiple comparisons test; only significant differences are shown; p1, p2, p3, p4, p5—the differences with HD, AIS, DM, CCCI, and SIVD, respectively—were significant according to Dunn’s multiple comparisons test.

**Table 9 ijms-25-01888-t009:** The relative (%, the percentage within total CD3+ and CD4+ cells) and absolute (#, the number of cells per 1 μL of whole peripheral blood) numbers of follicular Th-cell subsets in the patient groups.

Follicular Th Cell Subsets		HD (*n* = 17)	AIS (*n* = 24)	DM (*n* = 20)	CCCI (*n* = 33)	SIVD (*n* = 14)
Tfh1, CXCR3 + CCR6–	%	2.55 (2.01; 3.64)	2.36 (1.77; 3.66)	2.35 (1.88; 4.08)	2.06 (1.74; 3.16)	3.04 (2.11; 3.66)
	#	26 (23; 29)	28 (16; 41)	29 (21; 36)	24 (13; 38)	25 (19; 44)
Th2, CXCR3–CCR6–	%	2.24 (1.81; 3.35)	2.25 (1.68; 3.12)	2.37 (1.67; 2.78)	2.06 (1.64; 2.91)	2.73 (2.24; 3.45)
	#	25 (20; 35)	26 (16; 36)	22 (18; 31)	20 (13; 35)	21 (16; 41)
Tfh17, CXCR3–CCR6+	%	2.63 (1.85; 4.39)	3.61 (2.81; 5.17)	3.62 (2.56; 4.96)	2.89 (2.11; 3.97)	3.64 (2.81; 5.16)
	#	30 (20; 43)	39 (25; 55)	38 (28; 50)	28 (17; 45)	34 (25; 45)
DP Tfh, CXCR3 + CCR6+	%	1.04 (0.69; 1.52)	1.66 (1.21; 2.30)	1.61 (0.98; 2.80)	1.27 (0.98; 1.66)	1.41 (1.12; 2.50)
	#	10 (8; 14)	17 (11; 26)	17 (11; 30)	12 (7; 22)	14 (8; 20)

Note: The quantitative data (relative and absolute numbers of cells) are presented as median and quartile ranges (Med (Q25; Q75). The statistical analysis was performed with the Kruskal–Wallis test and Dunn’s multiple comparisons test; only significant differences are shown; p1, p2, p3, p4, p5—the differences with HD, AIS, DM, CCCI, and SIVD, respectively—were significant according to Dunn’s multiple comparisons test.

**Table 10 ijms-25-01888-t010:** Specific and non-specific biomarkers of the patient groups, compared to the control HD group.

	AIS	DM	CCCI	SIVD
Socio-demographic scores and cognitive status, non-specific	MOCA ↓	MOCA ↓	MOCA ↓	**MOCA ↓↓**
	MMSE ↓	MMSE ↓	MMSE ↓	**MMSE ↓↓**
			Life satisfaction ↓	**Life satisfaction ↓↓**
Socio-demographic scores and cognitive status, specific				Alcohol ↓
				Socialization ↓
				Physical activity ↓
Biochemical non-specific	Uric acid ↑	**Uric acid ↑↑**	**a1-AGP ↑↑**	Uric acid ↑
	Glyc ↑	a1-AGP ↑	Glyc ↑	**ALB ↓↓**
	VWF ↑	**Glyc ↑↑**	ALB ↓	**VWF ↑↑**
	ADAMTS13 ↓	VWF ↑	VWF ↑	**ADAMTS13 ↓↓**
Biochemical specific	NEFA ↑	Glu ↑		Glu ↓
		GGT ↑		Total Protein ↓
		TRIGS ↑		HDL ↓
				Chol ↓
				Fe ↓
EVs non-specific		**CD144 ↓↓**	CD144 ↓	
		**CD34 ↓↓**	CD34 ↓	
EVs specific	CD31 ↑			
	CD147 ↑			
Immunological non-specific	Th17-like cells, % ↑	**Th17-like cells, % ↑↑**		DP Th17, % ↑
	DP Th17, % ↑	**DP Th17, % ↑↑**		
	DP Th17, # ↑	**DP Th17, # ↑↑**		
Immunological specific		Th17-like cells, # ↑		Bm2′, # ↓
		Th17.1, % ↑		Bm3 + Bm4, % ↑
		Th17.1, # ↑		Plasmablasts, % ↑
				Plasmablasts, # ↑

Note: %—relative number; #—absolute number; ↑—statistically significant increase of the level; ↑↑—greatest statistically significant increase of the level among non-specific parameters (**also accompanied by bold font**); ↓— statistically significant decrease of the level; ↓↓—greatest statistically significant decrease of the level among non-specific parameters (**also accompanied by bold font**).

## Data Availability

The data presented in this study are available from the corresponding authors upon reasonable request. Data is not publicly available due to privacy.

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
