# Peer review of "Immunological Profile and Markers of Endothelial Dysfunction in Elderly Patients with Cognitive Impairments"

_ijms, 2024, doi:10.3390/ijms25031888_

Round 1

Reviewer 1 Report

Comments and Suggestions for Authors

The purpose of this study was immunological and biochem-ical profiling of elderly people with acute ischemic stroke (AIS), chronic cerebral circulation insuf-ficiency (CCCI), prediabetes and newly diagnosed type II diabetes mellitus (DM), and subcortical ischemic vascular dementia (SIVD). In general, the article is detailed and the design grouping is reasonable, the author believes that the clinical biochemical indicators are not good for the early diagnosis of senile diseases, and the comprehensive perspectives such as phenotypic markers and immune cell proportions are proposed to participate in the prediction, but they can be modified through the following points:

Major comment:

1.    This article mainly collected chronic diseases such as AIS/DM/CCCI/SIVD that are difficult to predict in the early stage. Authors can increase the comparison of some risk factors other than clinical indicators, such as smoking, obesity and other information, which can better illustrate the uncertainty of early diagnosis.

2.    Some early predictors of diseases, such as FPG, P2hPG, and HbA1c in type 2 diabetes, have become good early predictors. Other indicators, such as OGTT 1h indicators, should be further explained in the text.

Minor comments:

1.    It is more intuitive to use specific groups instead of numbers for the icon of Fig1, and the group indentation in the table will be ambiguous and should need to be changed.

2.    The abstract section should be further summarized to highlight the main core issues.

3.    The statistical analysis used Mann-Whitney U test, and other relevant analysis method should be mentioned in the text. For Mann-Whitney U test, larger sample sizes are necessary, but some groups have not enough samples to perform Mann-Whitney U test. Please check it. Wilcoxon rank sum test may be the best choice.

4.    The methods can be described in more detail. The whole manuscript should be carefully checked and edited before publication.

Comments on the Quality of English Language

The whole manuscript should be carefully checked and edited before publication.

Reviewer 2 Report

Comments and Suggestions for Authors

The manuscript examined the biomarkers of endothelial dysfunctions and immunological changes in elderly patients with cognitive dysfunction due to different underlying diseases. The manuscript itself is well written and explains the aim of the study with sufficient citations. However, there are several concerns about the current state of this manuscript.

Table 1: It should be named as Demographics of the participants and should include more information about the participants. Please refer to “https://www.ncbi.nlm.nih.gov/pmc/articles/PMC9931079/”. Furthermore, It would be informative to run statistical tests for the demographics.

Table 2: This table looks very confusing and should be edited in a way that reader can understand the information provided in the table. Furthermore, p values should be included in the table rather than providing just the asterisk. The choice of the statistical test for the diagnostics indicators for the different groups was Mann-Whitney test. However, Mann-Whitney test is not suitable for analyzing more than 2 groups and the authors should use Kruskall Wallis or anova for the comparison of the groups. Comparison of the disease group with each other is also important to provide information for the differentiation of different diseases. 

Futhermore, there are no evidence in the current version of the manuscript that authors used Kruskal-wallis test.

The source of the EVs and such details should also be mentioned in the results section to inform readers. I am aware that the details are in the methods section but while reading the results, the reader shouldn’t go back and forth with methods section all the time. Furthermore, there are no experiment of characterization of such EVs throughout the manuscript. 

The authors mentioned that EV flow cytometry controls were carried out according to the cited paper [101]. However, there are no evidence in the current version of the manuscript that the authors completed all required controls for the experiments for the EVs. The authors should also refer the article : https://www.tandfonline.com/doi/full/10.1080/20013078.2020.1713526  

The authors define SIVD without paraphrasing the article that they cited. They should do better job on paraphrasing at least. 

The discussion section is unnecessarily long and includes amyloid cascade, alzheimer’s disease etc which are not focus of this manuscript. 

Round 2

Reviewer 2 Report

Comments and Suggestions for Authors

I am satisfied with the authors' responses and changes in the manuscript.